# Sarcopenia Is Associated with an Increased Risk of Postoperative Complications Following Total Hip Arthroplasty for Osteoarthritis

**DOI:** 10.3390/biology12020295

**Published:** 2023-02-13

**Authors:** Kenny Chang, J. Alex Albright, Edward J. Testa, Alanna B. Balboni, Alan H. Daniels, Eric Cohen

**Affiliations:** 1Warren Alpert Medical School of Brown University, Providence, RI 02903, USA; 2Department of Orthopaedic Surgery, Warren Alpert Medical School of Brown University, Providence, RI 02903, USA

**Keywords:** hip, arthroplasty, sarcopenia, complications, readmission

## Abstract

**Simple Summary:**

Sarcopenia is an age-related disease characterized by uncontrolled muscle wasting and weakness in geriatric populations. Similarly, osteoarthritis is a prevalent condition involving irreversible joint damage. Late-stage osteoarthritis of the hip often requires surgical treatment with total hip replacement. Unfortunately, it is not uncommon for patients presenting for total hip replacement to also have underlying sarcopenia. Given the role of sarcopenia in musculoskeletal biology, as well as overall health, sarcopenia may affect the recovery and success of operations. However, the impact of sarcopenia on total hip replacement outcomes has not been well-defined. We have shown, in this study, that a diagnosis of sarcopenia prior to total hip replacement surgery is associated with significantly poorer postoperative outcomes as well as greater costs of care. This study identified an important risk factor for orthopedic surgeons to consider when treating elderly patients for late-stage osteoarthritis. Considering the postoperative risks, orthopedic surgeons may potentially opt to treat or reduce the severity of sarcopenia prior to surgery.

**Abstract:**

Sarcopenia is a state of catabolic muscle wasting prevalent in geriatric patients. Likewise, osteoarthritis is an age-related musculoskeletal disease affecting patients with similar demographics. Late-stage hip osteoarthritis is often treated with total hip arthroplasty (THA). As sarcopenia influences the surgical outcomes, this study aimed to assess the impact of sarcopenia on the outcomes of THA. A 1:3 matched case–control study of sarcopenic to control patients was performed using a large national database. In total, 3992 patients were analyzed. Sarcopenic patients undergoing THA were more likely to experience dislocation (odds ratio (OR) = 2.19, 95% confidence interval (CI) 1.21–3.91) within 1 year of THA. Furthermore, sarcopenic patients had higher urinary tract infection rates (OR = 1.79, CI 1.32–2.42) and a greater risk of 90-day hospital readmission (hazard ratio (HR) = 1.39, CI 1.10–1.77). Sarcopenic patients experienced more falls (OR = 1.62, CI 1.10–2.39) and fragility fractures (OR = 1.77, CI 1.34–2.31). Similarly, sarcopenic patients had higher day of surgery costs (USD 13,534 vs. USD 10,504) and 90-day costs (USD 17,139 vs. USD 13,394) compared with the controls. Ultimately, sarcopenic patients undergoing THA experience higher rates of postoperative complications and incur greater medical costs. Given the potential risks, orthopedic surgeons may consider treating or reducing the severity of sarcopenia before surgery.

## 1. Introduction

Sarcopenia is a widespread geriatric disease marked by a gradual decline in skeletal muscle mass and is further characterized by the replacement of muscle fibers with fat, changes in muscle metabolism, and a decrease in the quantity and quality of motor neurons [1]. In 2016, sarcopenia was assigned its own entry in the International Classification of Diseases Code, 10th Revision (ICD-10), signifying its recognition in the medical community as an independent disease state [2]. Sarcopenia is diagnosed using the strength, assistance with walking, rising from a chair, and falls (SARC-F) questionnaire, as well as physical tests and imaging [3]. The prevalence of sarcopenia is highly variable and ranges from 5% to 50% of older adults above the age of 60, especially those who reside in settings such as nursing homes and community dwellings, underscoring its relevance in orthopedic conditions affecting populations of a similar age group [1,4]. Even with conservative calculations, sarcopenia was estimated to have affected at least 50 million people in 2018 and is projected to affect over 200 million in the next 40 years because of the growing elderly population [5]. Although sarcopenia has been established as a risk factor affecting outcomes of cardiovascular, oncological, and gastrointestinal surgeries, there is a scarcity of research on the impacts of sarcopenia on common orthopedic procedures [6,7,8]. 

Similarly, osteoarthritis is one of the most common joint diseases affecting elderly individuals and has been ranked as the 10th leading contributor to disability [9]. Late-stage osteoarthritis of the hip is irreversible and is often treated with total hip arthroplasty (THA) [10]. Because of the demand from a continually growing elderly population, THA has become one of the most widely performed procedures [10]. By 2030, the annual number of primary hip arthroplasties performed is projected to grow by 174% [11]. However, despite the advancements in arthroplasty techniques and technologies, the number of implant-related complications and subsequent revision surgeries is expected to grow similarly [10,12]. For instance, THA revision is projected to grow by 137% from 2005 to 2030 [11]. Complications and revision surgeries are associated with significantly increased rates of short-term and long-term morbidity and mortality, as well as decreased physical well-being and quality of life [13,14]. In addition, THA revisions place a significant economic burden on the healthcare system, accounting for 19% of Medicare hip replacement expenditure [15]. As a result, it is essential to understand the potentially modifiable risk factors associated with adverse THA outcomes and revision surgeries. 

Patients suffering from sarcopenia are of a similar age group to those undergoing hip arthroplasty [16]. As a result, the similarity of these patient populations raises concerns regarding the effects of sarcopenia on the outcomes of THA. Currently, there are limited studies investigating this relationship. In a single-center study, Babu et al. demonstrated that sarcopenia is associated with periprosthetic joint infection following total joint arthroplasty, including both THA and total knee arthroplasty (TKA) procedures [17]. Despite its limitation of a small sample size (30 patients), Babu et al. offered a preliminary study indicating the need for a greater understanding of sarcopenia and age-related orthopedic procedures [17]. 

In this study, we aimed to further the understanding of the relationship between sarcopenia and THA. We used population-level data to quantify the impacts of sarcopenia on short-term postoperative outcomes after primary THA. We hypothesized that sarcopenia would be associated with increased rates of implant-related complications within 1 and 2 years of THA, as well as increased rates of 90-day medical complications. As a result, we also hypothesized that patients with sarcopenia undergoing THA are at greater risk of 90-day and 1-year all-cause hospital readmission, translating to increased costs of care. Lastly, we hypothesized that sarcopenic patients are more susceptible to falls and experience more fragility fractures following THA at the 1- and 2-year marks. 

## 2. Materials and Methods

### 2.1. Data Source

A retrospective analysis was performed using the PearlDiver (PearlDiver Technologies, Inc., Warsaw, IN, USA) database to query for de-identified data within the MHip subset of the larger Mariner dataset from 1 January 2012 to 31 October 2019. Generated using nationwide Humana Incorporated insurance claims, PearlDiver contains the records of over 120 million patients and allows for longitudinal tracking and analysis of the patients, procedures, medical diagnoses, and prescription medications. In this study, the longitudinal dataset was used to compare the rates of various postoperative complications among patients diagnosed with sarcopenia who underwent THA with a matched control population. As the PearlDiver patient records are de-identified, the study was exempt from approval by our institution’s review board. 

### 2.2. Creating the Experimental Cohort

Patients who underwent primary THA between 1 January 2012 and 31 October 2019 were identified using the Current Procedural Terminology (CPT), and the International Classification of Diseases, Ninth Revision (ICD-9) and 10th Revision (ICD-10) procedure codes. This timeframe allowed for a minimum of 2 years of follow-up data for each patient. Only patients active for at least 2 years before and 2 years after their THA were included to ensure complete medical records. Patients who previously underwent revision hip arthroplasty or underwent a contralateral total hip arthroplasty within 2 years of the first were excluded from the study. Additionally, patients with a history of multiple myeloma, Paget’s disease of the bone, metastatic cancer, achalasia, inflammatory bowel disease, cachexia, connective tissue/rheumatologic disease, epilepsy, poliomyelitis, myasthenia gravis, Parkinson’s disease, or bariatric surgery were also excluded, as these conditions can alter the metabolism of calcium or confound the relationship between sarcopenia and post-THA complications. For those interested in the complete list of CPT, ICD-9, and ICD-10 codes used, please contact the corresponding author.

The presence of sarcopenia was determined using ICD-9 (7282, muscle wasting and atrophy, not elsewhere classified, unspecified site) and ICD-10 (M6284, sarcopenia) codes. Patients with an insurance claim containing an associated diagnosis of sarcopenia within the 2 years before their THA were included in the “sarcopenia” group (the experimental group). The use of ICD-9 7282 to identify sarcopenia before the advent of ICD-10 M6284 is a previously established methodology [18,19]. Following identification, 1:3 matching with a control group was performed to account for age, sex, Charlson comorbidity index (CCI), tobacco use, diabetes, osteoporosis, osteoarthritis, chronic kidney disease, Vitamin D deficiency, and hyperparathyroidism. The procedure used for the design of the experimental cohort is shown in Figure 1. 

### 2.3. Determining and Comparing the Rates of Postoperative Complications

Through use of the CPT, ICD-9, and ICD-10 codes, the 1- and 2-year rates of implant-related complications, falls, and fragility fractures were calculated. The rates of various medical complications were calculated at 90 days. In addition, hospital readmission rates were assessed at the 90-day and 1-year marks. Costs of care during the day of surgery and at 90 days postoperatively were also extracted. The 90-day total costs included the costs derived from all services, readmissions, and medications. 

### 2.4. Statistical Analysis

*T*-tests and chi-square analyses were used to compare the characteristics of the experimental and control groups before and after the matching process. Multivariable logistic regression was used to compare the rates of postoperative complications between the matched experimental and control cohorts while controlling for tobacco use, diabetes, osteoporosis, chronic kidney disease, obesity, morbid obesity, Vitamin D deficiency, and hyperparathyroidism. Odds ratios (OR) and 95% confidence intervals (CIs) were calculated for each comparison. Kaplan–Meier failure and Cox regression analyses were used to assess the readmission rates. Student’s *t*-test was used to compare the costs of care. A *p*-value < 0.05 was used to determine statistical significance. All statistical analyses were performed using the R statistical package (v4.2.1; R Core Team 2022, Vienna, Austria) embedded within PearlDiver. 

## 3. Results

In total, 307,678 patients who underwent THA met the inclusion criteria. Of these, 1319 had a previous diagnosis of sarcopenia (0.43%). After the 1:3 matching process, 1014 patients were included in the experimental group and 2978 patients were included in the control group for further analysis (Figure 1). Table 1 shows the comparisons of the characteristics between the two cohorts before and after the matching process. After matching, there were no significant differences between the two cohorts.

### 3.1. Implant-Related Complications

Patients with sarcopenia undergoing primary THA were 70% more likely to experience implant-related complications one year after the operation. Patients with sarcopenia undergoing THA were 119% more likely to experience dislocation at 1-year follow-up but were not significantly more likely to experience a periprosthetic fracture, all-cause revision, prosthesis loosening, or deep periprosthetic infection. All ORs, 95% CI, and *p*-values are given in Table 2.

### 3.2. Ninety-Day Medical Complications and Readmission

Following THA, patients with sarcopenia were 37% more likely to experience 90-day medical complications than patients without sarcopenia. Specifically, at 90 days after THA, patients with sarcopenia were 79% more likely to experience a urinary tract infection (UTI) and 39% more likely to be readmitted than patients without sarcopenia (Figure 2). When evaluated at 1 year following THA, patients with sarcopenia were 55% more likely to have been readmitted to the hospital than patients without sarcopenia. All ORs, 95% CIs, and *p*-values are given in Table 3.

### 3.3. Total Costs of Care

When assessed on the day of surgery and 90 days postoperatively, patients with sarcopenia undergoing THA had higher costs of care than patients without sarcopenia. Patients with sarcopenia averaged USD 13,534.02 ± 14,816.43 on the day of surgery, while patients without sarcopenia averaged USD 10,504.32 ± 13,411.43 (*p* < 0.001): a 29% increase in the day of surgery costs for patients with sarcopenia. Ninety days postoperatively, patients with sarcopenia averaged USD 17,138.90 ± 18,781.20, while patients without sarcopenia averaged USD 13,393.82 ± 16,556.93 (*p* < 0.001), representing a 28% increase in 90-day total costs. 

### 3.4. Fall and Fragility Fracture Risks

Patients with sarcopenia undergoing THA were 62% more likely to experience a fall and 77% more likely to sustain a fragility fracture at 1-year follow-up. At 2 years, patients in the sarcopenia cohort were 70% more likely to experience a fall and 58% more likely to sustain a fragility fracture. All ORs, 95% CIs, and *p*-values are shown in Table 2.

## 4. Discussion

The results of this study support our primary hypothesis that patients with an insurance claim containing an associated diagnosis of sarcopenia within the two years before undergoing primary THA experience increased rates of postoperative complications, including implant dislocations, falls, fragility fractures, and UTIs, compared with a matched cohort of patients without a diagnosis of sarcopenia. Furthermore, these patients were more likely to require hospital readmission and experienced increased costs of care. This study adds to the growing body of literature on sarcopenia, as it clearly defines the association between sarcopenia and the rates of various orthopedic and medical complications, hospital readmission rates, and costs of care. This study has identified a crucial medical comorbidity for orthopedic surgeons to risk-stratify and optimize their patients before THA. 

Prior studies have found that sarcopenia is associated with poor functional outcomes following THA [20,21,22]. In particular, patients with sarcopenia demonstrate a slower gait speed, decreased hip abductor strength, lower hand grip strength, and reduced overall skeletal muscle mass following THA [20,21,22]. Combined with studies regarding the functional outcomes, the surgical complications presented in this current study emphasize the need to measure and improve the patient’s muscle and soft tissue health before surgery. Several studies have found the PSOAS:lumbar vertebral index (PLVI), calculated using computed tomography scans of the abdomen, to be a reliable measure of sarcopenia and a strong predictor of complications following various orthopedic procedures [17,23,24,25]. Babu et al. found that patients with sarcopenia, indicated by a PLVI lower than 0.842, were 257% more likely to experience a periprosthetic joint infection following THA or TKA [17]. The results of the present study, however, did not demonstrate that patients with sarcopenia were significantly more likely to experience a periprosthetic infection (*p* = 0.055) at the predetermined significance level, with our results falling just short of reaching statistical significance. Because the diagnostic methods for sarcopenia were not controlled for in the database used in this study, the severity of a patient’s sarcopenia with respect to PLVI could not be determined and thus may have contributed to the differing results. In addition, the findings by Babu et al. were derived from a sample of patients undergoing either THA or TKA [17]. As such, the inclusion of patients undergoing TKA may limit a direct comparison with the results of this present study [17]. 

Regarding other implant-related complications, our results demonstrated that patients with sarcopenia are approximately 100% more likely to experience a dislocation. Patients with sarcopenia may have inadequate peri-implant soft tissue, which may place them at a significant risk of dislocation. Prior studies have established that soft tissue tension is a key factor influencing implant instability and subsequent THA dislocation [26,27,28,29,30]. In a biomechanical study, Ogawa et al. found that patients who suffered from recurrent dislocation after primary total hip arthroplasty displayed a soft tissue tension approximately four times lower than patients who exhibited no dislocations [28]. Soft tissue tension in THA is a function of the strength and integrity of the surrounding hip musculature, such as the iliopsoas, hamstring, gluteal muscles, and quadriceps [26,27,28,29,30]. Unfortunately, patients with sarcopenia may have inadequate soft tissue tension because of their decreased skeletal muscle mass, which is particularly notable in the peri-implant musculature, often manifesting as short external rotators and gluteal muscles [27,31]. Ultimately, poor muscle quality or fatty infiltration followed by surgical trauma may accentuate the already inferior soft tissue status in the sarcopenic patient, further contributing to postoperative instability of the prosthesis [32]. Implant-related complications, such as dislocation and loosening, are not only strong predictors for revision arthroplasty but also poor patient-reported outcomes [12,33,34]. Hermansen et al. found that patients experiencing one hip dislocation reported markedly decreased hip-related and health-related quality of life and activity levels, and this impact persisted for up to 5 years [34]. When evaluated collectively, the present study provides a valuable addition to the risk factors associated with poor implant-related outcomes, as it strongly highlights the association of sarcopenia with THA dislocation rates. 

In addition, the overall skeletal muscle weakness and the impaired peri-implant soft tissue tension may predispose sarcopenic patients to experience higher rates of postoperative falls and fragility fractures, as reported in this study. As discussed previously, sarcopenic patients display poor functional outcomes following THA, including reduced gait speed and hip abductor strength [20,21,22]. Reduced gait speed and hip abductor strength indicate the loss of stable coordination, which may result in falls and subsequent fractures [35,36]. Furthermore, the soft tissue’s weakness may potentially result in the displacement of the femoral head from its optimal position and impingement during motion [30]. These changes may aggravate the weakness of the hip musculature and further contribute to potential falls and fractures. The increased risk of falls and fragility fractures is particularly concerning in this high-risk arthroplasty population. The stress of recovering from a THA in the sarcopenic patient, compounded by an additional fall and/or fragility fracture, may not only reduce the quality-of-life benefits usually provided by a THA but may also increase patient mortality [37].

Furthermore, the current study demonstrated that patients with sarcopenia undergoing THA are 79% more likely to experience postoperative medical complications such as UTIs. Similarly, Ardeljan et al. reported that patients with sarcopenia undergoing TKA were 64% more likely to experience 90-day medical complications, including urinary tract infections, than patients without sarcopenia [18]. Han et al. and Albright et al. reported similar findings [19,38]. Following THA, patients are often treated with indwelling catheterization to prevent postoperative urinary retention [39]. However, using catheterization for long periods increases the amount of residual urine and bacterial colonization at the catheter’s site, leading to a UTI [40]. Majima et al. found that elderly male patients with sarcopenia displayed impaired bladder contractility of the detrusor muscle [41]. The anatomical dysfunction of the bladder may additionally contribute to increased residual urine accumulation, predisposing the sarcopenic patient to a greater risk of UTI following THA. In addition, patients with sarcopenia generally have increased frailty, which may result in longer durations of postoperative immobilization and thus greater usage of catheterization [40]. Ultimately, this study’s findings of greater medical complications align with those of previous studies in orthopedics and other specialties [18,38,42,43]. 

As a result of the additional medical complications, sarcopenia is also associated with an extra USD 1.1 billion in healthcare costs [38,44]. In this study, we demonstrated that patients with sarcopenia undergoing THA have approximately 30% higher day of surgery and total 90-day costs, on par with the general finding that sarcopenia is linked to increased healthcare costs. These costs may be attributed to the specific surgical and medical complications discussed in this study, increased hospital stays, hospital readmissions, or more expensive hospital dispositions [33,45]. The finding in this study that patients with sarcopenia undergoing THA are at a greater risk of hospital readmissions at the 90-day and 1-year intervals further reflects the higher rates of postoperative complications and healthcare costs observed in this study. These findings point to the impetus of improving the health status of sarcopenic patients before total joint arthroplasty, not only to reduce the cost of care but, most importantly, to provide a possible reduction in both medical and surgical complications. 

Fortunately, treatments for sarcopenia exist, making the condition a potentially modifiable risk factor [46,47,48,49]. Current treatments for sarcopenia are focused on addressing the loss of muscle mass through a combination of physical exercise and nutritional interventions [46,47,48,49]. A randomized control trial showed that elderly patients diagnosed with sarcopenia and treated with a supplement of whey protein, leucine, and Vitamin D demonstrated an increase in gait speed compared with a group that received only dietary advice to increase their protein intake [50]. In another randomized control trial, a protein-enriched diet equivalent to 1.3 g/kg/day enhanced the effects of resistance training on increasing muscle mass and, subsequently, muscle strength [49]. Given that sarcopenia is a significant risk factor for poor postoperative outcomes and higher costs of care, orthopedic surgeons may consider risk-stratifying patients and recommending treatments for sarcopenia before THA. Future research should use a randomized controlled trial to investigate the effect of a preoperative protein supplementation and exercise treatment on implant-related and medical complications following THA.

This retrospective case–control study has several limitations. First, in line with the inherent limitations of retrospective studies, the accuracy of the data depends on the proper coding of items such as the CPT and ICD codes by administrators and physicians. Next, because of the nature of this dataset, we were unable to assess the severity of the patients’ sarcopenia, which may play a significant role in medical and surgical management. Similarly, we could not obtain and evaluate patient-reported and radiographic outcomes or identify the specifics regarding the laterality of the reported complications. However, we excluded patients with prior primary THAs; as such, this method effectively alleviated this potential issue with the dataset [51,52,53]. 

In addition, the diagnosis of sarcopenia may differ among providers, contributing to the variability in sarcopenia diagnoses and potential sampling bias. Furthermore, the prevalence of sarcopenia within this database (0.43%) is low compared with the prevalence described in previous literature, which may reflect the underdiagnosis of sarcopenia in orthopedics [1]. Moreover, the sarcopenic group only included patients with an insurance claim containing an associated diagnosis of sarcopenia within the two years before their THA. While this includes those patients diagnosed before the two-year period who subsequently had a claim with sarcopenia in their diagnosis list, there is the possibility of a patient subsequently dropping their diagnosis of sarcopenia before the two-year period and being left out of the analysis. As a result, some sarcopenic patients may be left out of the analysis entirely or may be within the control group because they were not captured by the coding in this database. If the control cohort contained some sarcopenic patients, the actual differences in the postoperative complication rates, hospital readmissions, and costs of care may be significantly larger than those described in this study. 

Additionally, because of the inability to determine the outcomes of dietary and nutritional counseling, we could not assess the impacts of a nutritional intervention using this database. Similarly, a physical exercise intervention could not be evaluated because of the lack of an osteoarthritis- or sarcopenia-specific indication. Lastly, although PearlDiver contains the health records of over 120 million patients, it is a private-payer insurance database. As a result, the external validity of the findings of this study may not hold for patients with different private or public insurance.

## 5. Conclusions

Patients with a diagnosis of sarcopenia within the two years before undergoing primary total hip arthroplasty are at a higher risk of experiencing implant-related and 90-day medical complications. In addition, patients with sarcopenia had higher readmission rates and costs of care and were also more likely to sustain a fall and have fragility fractures. These findings emphasize the clinical importance of sarcopenia to total hip arthroplasty in geriatric patients suffering from late-stage osteoarthritis. Further studies should evaluate the effectiveness of dietary protein supplementation and resistance training in reducing postoperative complications in patients with sarcopenia. 

## Figures and Tables

**Figure 1 biology-12-00295-f001:**
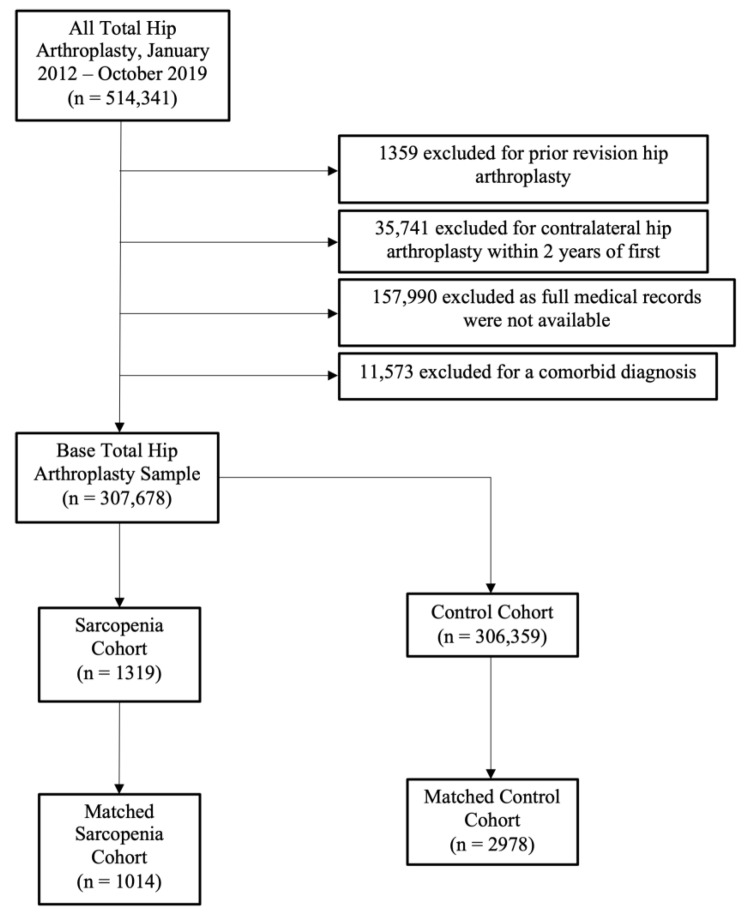
Diagram demonstrating the number of patients excluded from the study and the reasons why they were excluded.

**Figure 2 biology-12-00295-f002:**
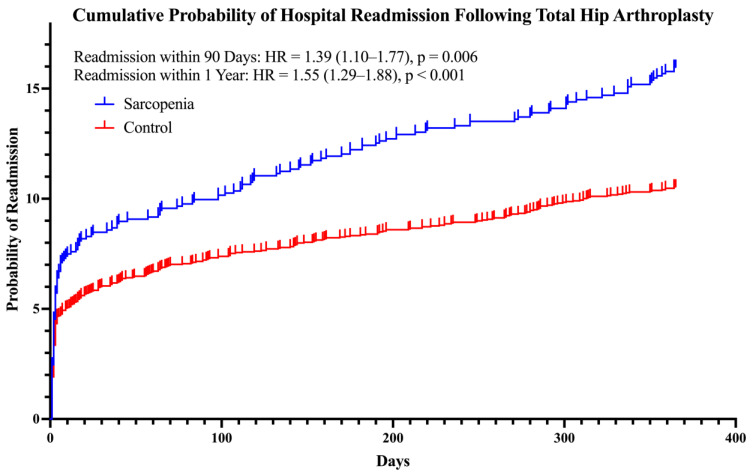
Kaplan–Meier failure analysis and Cox regression demonstrating the difference in cumulative hospital readmission rates following total hip arthroplasty for people with sarcopenia compared with the controls. Hazard ratios (HR) are reported with their respective 95% confidence intervals.

**Table 1 biology-12-00295-t001:** Characteristics of the sarcopenia and control cohorts before and after matching.

Characteristic	Unmatched			Matched		
	Sarcopenia (n = 1319)	Control (n = 306,359)	*p*-Value	Sarcopenia (n = 1014)	Control (n = 2978)	*p*-Value
Sex, female, n (%)	803 (60.9)	173,763 (56.7)	0.003	634 (62.5)	1862 (62.5)	1.000
Age, mean ± SD	65.4 ± 10.0	65.0 ± 10.0	0.086	66.4 ± 8.9	66.3 ± 8.9	0.935
CCI, mean ± SD	2.4 ± 2.7	1.5 ± 1.9	<0.001	1.6 ± 1.6	1.6 ± 1.6	0.472
Comorbidities, n (%)						
Osteoporosis	386 (29.3)	59,892(19.5)	<0.001	237 (23.4)	684 (23.0)	0.825
Osteoarthritis	1278 (96.9)	262,582(85.7)	<0.001	999 (98.5)	2933 (98.5)	1.000
Diabetes mellitus	667 (50.6)	121,346(39.6)	<0.001	482 (47.5)	1403 (47.1)	0.844
Tobacco use	744 (56.4)	119,292(38.9)	<0.001	415 (40.9)	1214 (40.8)	0.958
Obesity or overweight (BMI > 25)	626 (47.5)	125,541(41.0)	<0.001	442 (43.6)	1286 (43.2)	0.850
Morbid obesity (BMI > 40)	285 (21.6)	51,458(16.8)	<0.001	183 (18.0)	530 (17.8)	0.895
Vitamin D deficiency	507 (38.4)	91,886(30.0)	<0.001	346 (34.1)	995 (33.4)	0.707
Hyperparathyroidism	85 (6.4)	10,313(3.4)	<0.001	14 (1.4)	37 (1.2)	0.860

CCI = Charlson comorbidity index.

**Table 2 biology-12-00295-t002:** A comparison of the rates of short-term postoperative complications following total hip arthroplasty in patients with sarcopenia and a matched control.

Orthopedic-Related Complications, n (%)	Sarcopenia (n = 1014)	Control (n = 2978)	OR (95% CI)	*p*-Value
Total orthopedic-related complications				
1 year	48 (4.73)	90 (3.02)	1.70 (0.78–3.54)	0.012
2 years	51 (5.03)	106 (3.56)	1.67 (0.79–3.37)	0.042
Periprosthetic fracture (%)				
1 year	11 (1.08)	19 (0.64)	1.70 (0.78–3.54)	0.167
2 years	12 (1.18)	21 (0.71)	1.67 (0.79–3.37)	0.159
All-cause revision (%)				
1 year	32 (3.16)	63 (2.12)	1.51 (0.97–2.31)	0.063
2 years	35 (3.45)	77 (2.59)	1.34 (0.89–2.00)	0.154
Instability (%)				
1 year	20 (1.97)	27 (0.91)	2.19 (1.21–3.91)	0.009
2 years	21 (2.07)	31 (1.04)	2.00 (1.13–3.48)	0.015
Loosening (%)				
1 year	<11 (N/A)	12 (0.40)	2.20 (0.90–5.23)	0.075
2 years	11 (1.08)	16 (0.54)	2.01 (0.90–4.32)	0.076
Dislocation (%)				
1 year	20 (1.97)	27 (0.91)	2.19 (1.21–3.91)	0.008
2 years	20 (1.97)	30 (1.01)	1.97 (1.10–3.46)	0.020
Deep periprosthetic infection (%)				
6 months	16 (1.58)	29 (0.97)	1.63 (0.86–2.98)	0.122
1 year	19 (1.87)	32 (1.07)	1.75 (0.99–3.11)	0.055
Experienced a fall (%)				
1 year	42 (4.14)	77 (2.59)	1.62 (1.10–2.39)	0.014
2 years	68 (6.71)	121 (4.06)	1.70 (1.24–2.30)	<0.001
Fragility fracture (%)				
1 year	95 (9.37)	168 (5.64)	1.77 (1.34–2.31)	<0.001
2 years	105 (10.36)	206 (6.92)	1.58 (1.22–2.03)	<0.001

OR = odds ratio; CI = confidence interval.

**Table 3 biology-12-00295-t003:** A comparison of the rates of 90-day medical complications following total hip arthroplasty in patients with sarcopenia and the matched controls.

Medical Complications, n (%)	Sarcopenia (n = 1014)	Control (n = 2978)	OR (95% CI)	*p*-Value
Total 90-day medical complications	156 (15.38)	349 (11.72)	1.37 (1.11–1.69)	0.003
Acute kidney injury	22 (2.17)	67 (2.25)	0.94 (0.56–1.51)	0.790
Cardiac arrest	0 (0)	<11 (N/A)	N/A	0.994
Deep vein thrombosis	<11 (N/A)	<11 (N/A)	1.17 (0.37–3.75)	0.79
Wound disruption	<11 (N/A)	21 (0.71)	0.97 (0.39–2.17)	0.935
Hematoma	15 (1.47)	23 (0.77)	1.91 (0.97–3.65)	0.053
Nerve injury	<11 (N/A)	0 (0)	N/A	0.996
Pneumonia	16 (1.58)	50 (1.68)	0.92 (0.50–1.59)	0.768
Pulmonary embolism	11 (1.08)	21 (0.71)	1.54 (0.71–3.14)	0.249
Required transfusion	45 (4.44)	105 (3.53)	1.25 (0.87–1.79)	0.217
Urinary tract infection	74 (7.30)	127 (4.26)	1.79 (1.32–2.42)	<0.001

OR = odds ratio; CI = confidence interval.

## Data Availability

The PearlDiver database is a private national database that requires private access to a password-protected server.

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
