# Peer review of "Sarcopenia Is Associated with an Increased Risk of Postoperative Complications Following Total Hip Arthroplasty for Osteoarthritis"

_biology, 2023, doi:10.3390/biology12020295_

Round 1

Reviewer 1 Report

The article could be accepted with minor changes. I propose to review the following changes: 

In Materials and methods: In Data Source (line 87) the dataset is from 2010 and 2018. In Creating the Experimental Cohort (line 96) the dates are from January 2012 and october 2019. Dates do not match. 

In Materials and methods: last paragraph when talking about presence of sarcopenia it would be interesting to specify if  nutritional intervention was considered in case it was reported in the discharge reports. In case it was not considered it could be reflected in the limitations of the study.

In table 1 iclude:  Obesity or overweight (BMI 25). Morbid obesity (BMI

40)

In results: the authors could especify if the costs include only the cost attributable to readmissions or also the costs derived from medication

Author Response

Please see the attachment. Thank you for your time and effort. 

Reviewer 2 Report

1.      Line 10-21, simple summary? Please delete it and only leave abstract.

2.      Please end your abstract with a "take-home" message.

3.      It is unclear whether the author's something new in this work. According to evaluation, several published studies by other researchers in the past adequately explain the issues you made in the present paper. Please be careful to highlight in the introduction section anything really innovative in this work.

4.      In order to highlight the gaps in the literature that the most recent research aims to fill, it is crucial to review the benefits, novelty, and limitations of earlier studies in the introduction.

5.      Line 76-82, make it as narrative, not pint by point.

6.      Line 53-54, additional reference is needed to support the explanation regarding total hip arthroplasty. Furthermore, the MDPI's suggested reverence should be applied in the manner described below to further support this description as follows: Ammarullah, M. I.; Santoso, G.; Sugiharto, S.; Supriyono, T.; Wibowo, D. B.; Kurdi, O.; Tauviqirrahman, M.; Jamari, J. Minimizing Risk of Failure from Ceramic-on-Ceramic Total Hip Prosthesis by Selecting Ceramic Materials Based on Tresca Stress. Sustainability 2022, 14, 13413. https://doi.org/10.3390/su142013413

7.      To enhance the understandability of the section on materials and methods easier for them to understand rather than just depending on the main text as it exists at the moment, the authors could add additional illustrations in the form of figures that explain the workflow of the present study.

8.      The revised manuscript after peer review must provide detailed information on the error and tolerance of the experimental equipment utilized in this study. Due to the disparate outcomes of other researchers' subsequent studies, it would make for a valuable discussion.

9.      Outcomes must be compared to similar past research.

10.   The discussion in present article is extremely poor in quality as overall. The authors must elaborate on their arguments and provide a thorough justification. Don't just state the results and give a quick explanation.

11.   Please discuss the further research in the conclusion section.

12.   Literature from the last five years should be enriched to reference. MDPI reference is strongly recommended.

13.   The manuscript needs to be proofread by the authors since it has grammatical and language issues.

14.   Graphical abstract is encouraged to provide in submission after review.

Author Response

(The authors gave the same response as above.)

Reviewer 3 Report

The authors present a study looking at the relationship of sarcopenia and total hip replacement surgery outcomes.

The paper is nicely presented and succinct. 

Significant limitations came to mind while reading, these are somewhat addressed by the authors. I would expect there to be a massive sampling bias here. I am 100% certain that the control group would contain a number of sarcopenic patients. The sarcopenic group includes those diagnosed <2 years ago. There exist those with a diagnosis 2+ years, the conflict between accurate diagnosis and insurance record factors to also consider. 

Sarcopenia is described by the authors as a treatable condition, it would have been very interested to see if there was any difference between those with sarcopenia who are, and are not, engaging with active management of this condition. Showing that treatment could truly impact surgical outcome would have been a powerful conclusion.  Would this be possible to determine using the dataset? I understand you quote only prescribed medications are included in the dataset, but wonder if this would include prescribed dietary supplements or even physical therapy regimens?

Author Response

(The authors gave the same response as above.)

Round 2

Reviewer 2 Report

Reviewers greatly appreciate the efforts that have been made by the author to improve the quality of their articles after peer review. I reread the author's manuscript and further reviewed the changes made along with the responses from previous reviewers' comments. Unfortunately, the authors failed to make some of the substantial improvements they should have made making this article not of decent quality with biased, not cutting-edge updates on the research topic outlined. In addition, the author also failed to address the previous reviewer's comments, especially on comments number 3 (nothing really novel with cutting edge insight), 4 (not clear explanation), and 6 (suggested reference not incorporated). Thank you very much for the opportunity to read the author's current work.

Reviewer 3 Report

I believe that this revised edit of the paper is far superior to the initial submission. I appreciate the recognition of the limitations due to coding  and associated selection bias which are otherwise beyond control. The authors make succinct observations of the available data, and also acknowledge the need for further studies including role of nutrition and exercise in sarcopenia which is both very relevant and of wide interest to readers.